# Soil Organic Carbon Mapping from Remote Sensing: The Effect of Crop Residues

**Klara Dvorakova [1],\*, Pu Shi [1], Quentin Limbourg [2] and Bas van Wesemael [1]**

[1] Georges Lemaître Centre for Earth and Climate Research, Earth and Life Institute, Université Catholique de Louvain, 1348 Louvain-la-Neuve, Belgium; pu.shi@uclouvain.be (P.S.); bas.vanwesemael@uclouvain.be (B.v.W.)

[2] Walloon Agricultural Research Centre (CRA-W), Farming Systems, Territories and Information Technology Unit, 5030 Gembloux, Belgium; q.limbourg@cra.wallonie.be

\* Correspondence: klara.dvorakova@uclouvain.be

**Abstract:** Since the onset of agriculture, soils have lost their organic carbon to such an extent that the soil functions of many croplands are threatened. Hence, there is a strong demand for mapping and monitoring critical soil properties and in particular soil organic carbon (SOC). Pilot studies have demonstrated the potential for remote sensing techniques for SOC mapping in croplands. It has, however, been shown that the assessment of SOC may be hampered by the condition of the soil surface. While growing vegetation can be readily detected by means of the well-known Normalized Difference Vegetation Index (NDVI), the distinction between bare soil and crop residues is expressed in the shortwave infrared region (SWIR), which is only covered by two broad bands in Landsat or Sentinel-2 imagery. Here we tested the effect of thresholds for the Cellulose Absorption Index (CAI), on the performance of SOC prediction models for cropland soils. Airborne Prism Experiment (APEX) hyperspectral images covering an area of 240 km$^2$ in the Belgian Loam Belt were used together with a local soil dataset. We used the partial least square regression (PLSR) model to estimate the SOC content based on 104 georeferenced calibration samples (NDVI < 0.26), firstly without setting a CAI threshold, and obtained a satisfactory result (coefficient of determination ($R^2$) = 0.49, Ratio of Performance to Deviation (RPD) = 1.4 and Root Mean Square Error (RMSE) = 2.13 g kgC$^{-1}$ for cross-validation). However, a cross comparison of the estimated SOC values to grid-based measurements of SOC content within three fields revealed a systematic overestimation for fields with high residue cover. We then tested different CAI thresholds in order to mask pixels with high residue cover. The best model was obtained for a CAI threshold of 0.75 ($R^2$ = 0.59, RPD = 1.5 and RMSE = 1.75 g kgC$^{-1}$ for cross-validation). These results reveal that the purity of the pixels needs to be assessed aforehand in order to produce reliable SOC maps. The Normalized Burn Ratio (NBR2) index based on the SWIR bands of the MSI Sentinel 2 sensor extracted from images collected nine days before the APEX flight campaign correlates well with the CAI index of the APEX imagery. However, the NBR2 index calculated from Sentinel 2 images under moist conditions is poorly correlated with residue cover. This can be explained by the sensitivity of the NBR2 index to both soil moisture and residues.

**Keywords:** soil organic carbon mapping; cellulose absorption index; crop residues; hyperspectral data; partial least squares regression

## 1. Introduction

Soil organic carbon (SOC) is the major terrestrial carbon pool [1], with exchanges with atmospheric CO$_2$ [2]. Sanderman et al. found that the SOC pool in croplands is a result of a long history of cultivation, and thus they argued that croplands have a large potential to restore SOC and sequester

atmospheric $CO_2$ [3]. This will also lead to a gradual restoration of soil fertility and, in many cases, crop yield through nutrient use efficiency [3].

Optical and microwave remote sensing data have been used for soil carbon (C) stocks estimations mainly in cropland soils [4]. Most commonly, empirical or machine learning approaches combined with optical remote sensing of exposed soils are being used for estimating SOC. Optical remote sensing data are acquired in the visible (Vis), near-infrared (NIR) and shortwave infrared (SWIR) domain (VNIR-SWIR, 400–2500 nm). Preparing for an operational earth observation platform for soil monitoring, an increasing number of pilot studies have focused on the potential of optical remote sensing techniques in SOC estimation of bare soils [5–11]). These pilot studies are restricted to croplands where (i) soil properties are fairly uniform in the top 0–30 cm due to soil mixing during tillage, (ii) the soil is exposed and an eventual soil crust has been ploughed in before seeding, and (iii) surface water content is generally low during the cloud free sky required for useful remote sensing imagery [12]. The applications of satellite imagery for predicting soil properties are rapidly developing as multispectral imagery is already freely available from the Sentinel-2 (S-2) multi spectral instrument (MSI) and hyperspectral satellites such as Prisma [13] have been launched or are planned for the near future (EnMAP [14] or Shalom [15]). Spectroscopy in outdoor conditions is constrained by atmospheric conditions and spatial variation in surface soil properties. The prediction models assume that a specific soil property i.e., SOC influences the reflectance within the VNIR-SWIR (400–2500 nm) range. Therefore, other factors that also influence the reflectance spectrum will disturb the prediction of the soil chromophore in question [16]. The main factors affecting the soil reflectance spectrum are soil water content, vegetation, crop residues, and soil roughness [17]. On rough soils with aggregates of 20 to 25 mm, Rodionov et al. obtained an overestimation of 61% of the SOC content [18]. Nocita et al. obtained a poor SOC prediction when they did not account for soil moisture: an RMSE of 30.21 g kg$^{-1}$ and an $R^2$ of 0.25 when a calibration on dry samples was applied to moist samples [19]. Bartholomeus et al. overestimated SOC content by > 5 g kg$^{-1}$ at a fractional maize cover of 5% to 10% [20]. Rodionov et al. obtained an increase in SOC content by almost 30 g kg$^{-1}$ when 5% of the surface was covered with green leaves and straw [21]. Hence, if SOC is to be estimated over increasingly large areas, we must be able to verify, at a distance, that the soil reflectance spectrum is not affected by these disturbing factors.

The effect of crop residues on soil reflectance has been addressed in several studies. A lignocellulose absorption at 2100 nm in the reflectance spectra of crop residues has been observed due to structural compounds such as cellulose, hemicellulose and lignin [22]. Daughtry et al. defined a spectral variable, the cellulose absorption index (CAI), which is used for discriminating crop residues from soil [22]. The index is based on the depth of the lignocellulose absorption feature in the shortwave infrared region (2000–2200 nm). Nagler et al. found that CAI linearly increased for crop residues ranging from 0% (bare soil) to 100% cover and concluded that CAI is relevant to situations where it is important to distinguish residues from soils [23]. However, also the main compounds of soil organic matter (SOM; cellulose, lignin and starch) affect reflectance around 2100 and 2300 nm [24], and they might be masked by crop residue, as both SOM and residues impact the same spectral features. Rodionov et al. established a non-linear relationship between CAI and SOC overestimation in croplands, and proposed a correction of laboratory SOC predictions using the CAI as an indicator for crop residues [21].

The S-2 mission of the EU Copernicus program has shown to be promising for soil mapping, due to its high spatial resolution, free data availability and short revisit time. It consists of 13 bands including two SWIR bands (B11 and B12), which are particularly useful to exploit the spectral signature that is assigned to SOC content [24]. Castaldi et al. demonstrated that S-2 data can be successful in predicting SOC content in a relatively small pilot area restricted to bare cropland soils [8]. The main challenge, however, is the minimization of disturbing factors, such as residues and soil moisture on multispectral images. Demattê et al. applied different thresholds based on the The Normalized Burn Ratio (NBR2) index calculated on Landsat multispectral data in order to remove soil spectra affected by crop residues [25]. Castaldi et al. used the NBR2 index on S-2 multispectral data and tested the effect

of the NBR2 threshold on the performance of SOC prediction models [26]. However, the capability of the multispectral NBR2 index for crop residue detection and soil moisture quantification still has to be proven.

To the best of our knowledge, CAI has not yet been used for SOC estimation by means of remote sensing. Thus, our first objective is to test the effect of a CAI threshold on the performance of SOC prediction models using the Airborne Prism Experiment (APEX) hyperspectral sensor. We hypothesize that SOC prediction can be optimized by restricting the number of calibration samples by applying a CAI threshold. Our second objective is to establish whether the NBR2 calculated from the Sentinel 2 imagery can be used as a proxy for the CAI, as the latter can only be used for hyperspectral sensors.

## 2. Materials and Methods

### 2.1. Study Site and Sample Collection

The study was conducted in an area of 240 km² located in the northern part of Wallonia, Belgium (NW: 50°35′50′′N 4°39′28′′E; SE: 50°42′07′′N 5°06′21′′E; Figure 1). The area falls within the loam belt region dominated by niveo-eolian deposits, where well-drained soils are found. The relief is gently undulating with altitudes varying between 80 m (in the north-east) and 178 m (in the south-west). The climate is temperate oceanic with mean annual precipitation of 790 mm and with the lowest monthly mean temperature in January (2.3 °C) and the highest monthly mean temperature in July (17.8 °C) [7]. Predominant land use in the area is conventionally cultivated cropland with winter wheat, sugar beet, maize and potatoes as the dominant crops grown in a three-year rotation.

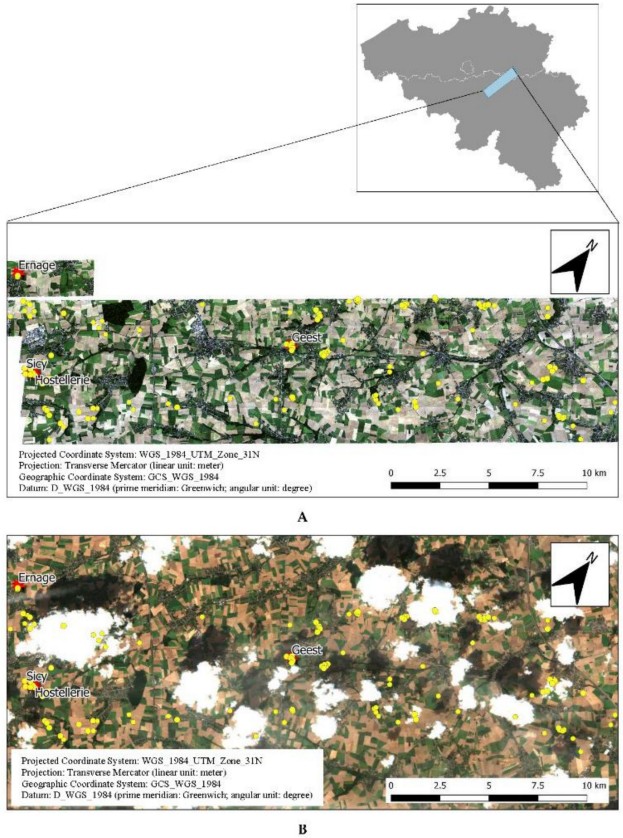

**Figure 1.** Location of the study area in the Belgian loam belt and RGB image acquired by (**A**) the hyperspectral Airborne Prism Experiment (APEX) sensor on 2 September 2018 (red: 616 nm, green: 551 nm, blue: 470 nm) and (**B**) the Multi-spectral Instrument (MSI) Sentinel-2 sensor on 24 August 2018 (red: 665 nm, green: 560 nm, blue: 490 nm). The yellow dots indicate the location of the sampling points, the red stars the location of the three extensively sampled fields.

A total of 104 surface soil samples (0–10 cm) were randomly collected in October 2018 and September 2019 (see yellow points on Figure 1) within fields that were without vegetation (selected based on a Normalized Difference Vegetation Index (NDVI) threshold, see further) during the APEX data acquisition (2 September 2018) [27]. These samples consisted of five sub-samples collected at random locations within a 1 m radius centered on the geographical position of a sampling plot, which was recorded by a Garmin GPS instrument with 3 m precision. The sub-samples were then thoroughly mixed and stored in a plastic bag. They were air-dried, gently crushed and passed through a 2 mm sieve. SOC was analyzed by dry combustion, using a VarioMax CN Analyzer (Elementar Analysensysteme GmbH, Hanau, Germany), as detailed in Shi et al. [27]. For samples showing clear reactions with 10% HCl, carbonate content was measured using a modified pressure-calcimeter method [28]. Then, SOC was obtained by subtracting the inorganic carbon content from the total carbon.

Additionally, an extra set of 276 soil samples covering three fields (Sicy, Hostellerie and Ernage; Figure 1) was obtained from the Walloon Agricultural Research Center (CRA-W) database. The surface soil samples (0–25 cm) were collected in August 2018 according to a regular grid (Figure 2). Each sample consisted of 10 sub-samples taken within a radius of 2.5 m and the latter were then thoroughly mixed for a representative sample. Sample positions were recorded using a John Deere Starfire 3000 Real Time Kinematic (RTK) GPS instrument with 2.5 cm precision. The SOC content was analyzed by means of the dry combustion method. As the two sets of samples were part of two separate campaigns, the sampling strategies were different. We, however, hypothesize that the difference in sampling depth between the two datasets of 104 samples and 276 samples (10 cm and 25 cm, respectively) do not induce a bias, as all fields are annually ploughed to a depth of c. 25 cm creating a uniform SOC content in the plough layer.

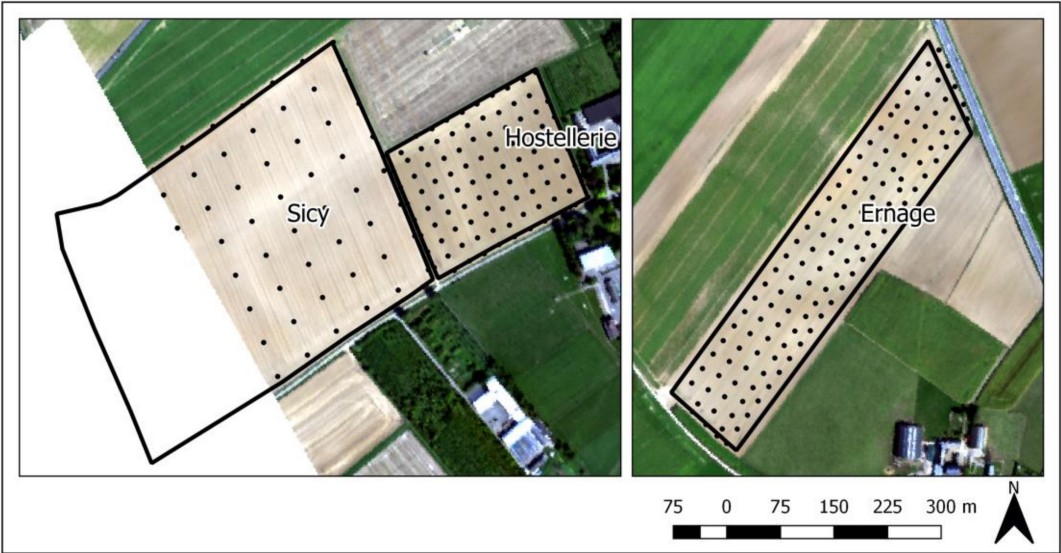

**Figure 2.** RGB image of the Sicy, Hostellerie and Ernage fields in the Belgian loam belt acquired by the hyperspectral Airborne Prism Experiment (APEX) sensor on 02 September 2018 (red: 616 nm, green: 551 nm, blue: 470 nm) and the locations of the soil samples.

*2.2. Remote Sensing Data*

2.2.1. Airborne Images

A joint Swiss–Belgian airborne imaging spectrometer (APEX) acquired hyperspectral imagery on 2 September 2018 between 10:44 and 11:56 local time in clear sky conditions (Figure 1A, Table 1). The APEX data were pre-processed at the VITO Remote Sensing Department, using the Central Data Processing Center (CDPC) for airborne Earth observation data, and are provided as georeferenced at-surface reflectance [29]. The radiometric spectral calibration was performed by means of calibration

cubes generated from data measured and collected on the APEX Calibration Home Base hosted at German Aerospace Centrum (DLR) Oberpfaffenhofen and the geometric correction was based on direct georeferencing [30,31]. Noisy or troublesome bands, i.e., the water bands (1310–1555 nm, 1750–2000 nm) and the first blue bands with poor radiometric quality [29] (413–440 nm) were removed. For more information on the APEX imaging spectrometer, you are kindly referred to Schaepman et al. [32].

**Table 1.** Comparison of the Multispectral Instrument (MSI) on board the Sentinel-2 constellation and the Airborne Prism Experiment (APEX) sensor's characteristics [29,33]. VNIR = visible and near-infrared, SWIR = shortwave infrared.

|  | **MSI** | **APEX** |
| --- | --- | --- |
| Altitude (km) | 786 | 3.6 |
| Sensor type | multispectral | hyperspectral |
| Spectral range (nm) | 443–2190 | 413–2431 |
| Spectral bands | 13 | 285 |
| Resolution |  |  |
| Spatial (m) | 10–20–60 | 2 |
| Temporal (day) | 5 | — |
| Spectral (nm) | 15–180 | 2–13 |
| Noisy bands (nm) | — | 413–440, 1310–1555, 1750–2000 |
| Signal to noise (SNR) |  |  |
| VNIR | 89:1 to 168:1 | 50:1 to 700:1 [8] |
| SWIR | 50:1 to 100:1 | 40:1 to 600:1 * [8] |

* After noisy band removal.

In order to assess the quality of the APEX hyperspectral data, spectral reflectance of one black and one white reference surface were acquired at the moment of the APEX data acquisition on 2 September 2018 by means of an ASD FieldSpec portable spectral radiometer (Analytical Spectral Device Inc., USA). The spectral resolution ranges from 3 nm (at 350–1000 nm) to 10 nm (at 1000–2500 nm), and the spectral readings are acquired in 1 nm increments over the wavelength range. The references were a black asphalt parking lot (50°34′35.43′′N; 4°42′14.7′′E) and a white marl square (50°35′55.9′′N; 4°50′06.7′′E). The geographical position of each sampling point was recorded by a Garmin GPS instrument with 3 m precision. The sensor was positioned from the nadir approximately 1 m above the ground and the average value of nine consecutive measurements for each reference was calculated as the final spectral reference. The spectral bands from 350 to 440 nm and from 2450 to 2500 nm were removed because of instrumental noise. For comparison with airborne data, the APEX reflectance spectra were extracted for the georeferenced black and white references, using the bilinear interpolation technique. The latter assigns the output cell value by taking the weighted average of four closest cell centers.

2.2.2. Spaceborne Images

Multispectral field reflectance data were obtained using the multispectral instrument (MSI) aboard the S-2 platform (Table 1). The images were provided as Level 2A product, i.e., geometrically, radiometrically and atmospherically corrected. Since the MSI has 13 bands with different spatial resolutions, the images were spatially resampled (nearest neighbor resampling) at 10 m to maximize the level of detail of the S-2 data.

Firstly, an S-2B image with minimum cloud cover acquired on 24 August 2018 was downloaded from the Copernicus open access hub (Figure 1B). The image was selected to coincide as much as possible with the APEX image. Clouds and their shadows on the S-2B image were masked using the Sentinel Application Platform (SNAP) software. The image was further geometrically registered using the airborne data as a reference image, to correct the geometrical shift of the S-2B image and thus obtain an overlap between S-2B and airborne images. Then, an S-2A image acquired on 13 October, 2019 was downloaded from the Copernicus open access hub, with the aim to test whether a link exists

between ground-truth crop residue cover observations and the reflectance in the SWIR bands of the S-2 MSI. This date was selected because the image was cloud-free and it was acquired in autumn when many fields were bare after the harvest of potatoes and sugar beets.

### 2.3. Spectral Indices

Firstly, an NDVI threshold was set to mask green vegetation (Table 2). The threshold was determined by (i) visually inspecting the RGB images from the dates of overflight (Figure 1) and by (ii) minimizing the "salt-and-pepper" patchiness of the resulting mask. Pixels with NDVI values below 0.26 were kept for APEX, below 0.30 for S-2 (September 2018) and below 0.35 for S-2 (October 2019). Then, the wavelengths used to calculate the CAI (Table 2) were somewhat adjusted, taking into account (i) the quality of the APEX airborne spectra and (ii) the theoretical positioning of the bands, which were defined by Daughtry to be centered at 2030, 2100 and 2210 nm [34]. Finally, an NBR2 index which is calculated as the normalized difference between B11 and B12 on S-2 MSI (Table 2) was used for exploiting the correlation with crop residue cover on S-2 multispectral data.

**Table 2.** Details of derived indices and the wavelengths (nm) used in this work for the Multispectral Instrument (MSI) on board the Sentinel-2 constellation and the Airborne Prism Experiment (APEX) sensor. R: reflectance of specific wavelength, NDVI: Normalized Difference Vegetation Index, CAI: cellulose absorption index, NBR2: Normalized Burn Ratio2.

| Index | Equation | MSI Bands | APEX Bands | Reference |
|---|---|---|---|---|
| NDVI | $NDVI = \frac{R_{NIR} - R_{red}}{R_{NIR} + R_{red}}$ | $R_{red}$: 665 nm (B4)<br>$R_{NIR}$: 842 nm (B8) | $R_{red}$: 664 nm<br>$R_{NIR}$: 842 nm | [35] |
| CAI | $CAI = 0.5\,(R_{2.0} + R_{2.2}) - R_{2.1}$ | | $R_{2.0}$: 2026 nm<br>$R_{2.1}$: 2100 nm<br>$R_{2.2}$: 2214 nm | [34] |
| NBR2 | $NBR2 = \frac{R_{SWIR1} - R_{SWIR2}}{R_{SWIR1} + R_{SWIR2}}$ | $R_{SWIR1}$: 1610 nm (B11)<br>$R_{SWIR2}$: 2190 nm (B12) | | [36] |

### 2.4. Weather Data

Agri4Cast weather data available for August 2018 recorded 74 mm precipitation for August 2018 (Figure 3A). Moreover, during the acquisition of S-2 and APEX images dry conditions prevailed, as a dry period of at least three days preceded each image acquisition date. Additionally, daily meteorological data from the Royal Meteorological Institute (RMI) weather station Sombreffe (50°31′46″N 4°36′03″E) were retrieved for October 2019 (Figure 3B). One rainy episode occurred on the day of the S-2 image acquisition (13 October 2019) in the early morning (personal observation). This heavy rainfall occurred before the S-2 image was acquired at 10:50 local time.

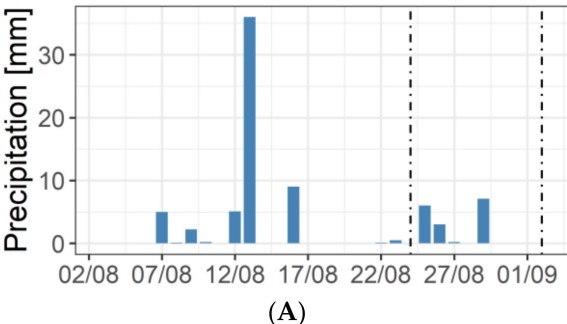

(A)

**Figure 3.** *Cont.*

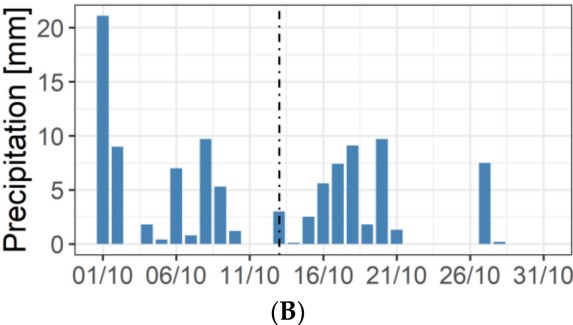

**(B)**

**Figure 3.** (**A**) Daily precipitation over the Belgian loam belt study area for August 2018 and the beginning of September 2018. The vertical dashed lines represent the acquisition dates of the Sentinel-2 (24 August) and Airborne Prism Experiment (APEX) images (2 September). Precipitation from weather stations was interpolated on 25 × 25 km grid. Source: Agri4Cast. (**B**) Daily precipitation recorded at the Sombreffe weather station in October 2019. The vertical dashed line represents the acquisition date of Sentinel-2 image on 13 October 2019. Source: CRA-W/Réseau Pameseb.

## 2.5. Soil Organic Carbon Predictive Models

### 2.5.1. Spectral Pre-Processing

VNIR-SWIR spectra were extracted from the APEX images, to which the NDVI mask has been applied, at the locations of the 104 soil samples by means of bilinear interpolation technique. This method assigns the output cell value by taking the weighted average of the four closest cell centers. The spectral matrix was passed through several pre-processing techniques, commonly used in soil spectroscopy with the aim to decrease physically related variations, remove noisy features and emphasize the important features along the spectrum [9,37]. The pre-treatments included: (i) absorbance (-log reflectance), (ii) 1st and 2nd order derivatives [38], (iii) standard normal variate transformation and detrending [39] (iv) continuum removal [40] and (v) Savitzky–Golay smoothing [41] with a window size of 20 nm and 2nd order polynomial and derivatives. In addition, all methods were calculated on (i) raw reflectance/absorbance spectra and (ii) Savitzky–Golay smoothed spectra. All pre-processing transformations were performed in "prospectr" package developed in R and the pre-treatment providing the model with the best performance was selected [42].

### 2.5.2. PLSR Model

The PLSR approach uses the full spectrum to establish a linear regression model where the significant spectral information is contained in a few orthogonal factors, called latent variables (LV) [43,44]. Because a limited number of samples were available, a ten-fold-cross-validation procedure was adopted to estimate the prediction capability of the PLSR model for the training set. The PLSR analyses were performed using the "pls" package developed in R software [42]. To avoid over- or under-fitting, the optimal number of LV was determined as the one producing a model having the minimal Root Mean Square Error (RMSE) of cross-validation, while the maximum number of LV possible was set to 15. In order to detect the main spectral regions involved in the SOC prediction, the Variance Importance Projection (VIP) index was calculated. Spectral bands with VIP values greater than one were considered important in the PLSR model.

The performance of the prediction models was assessed using the following parameters: coefficient of determination ($R^2$) (Equation (1)), Root Mean Square Error (RMSE) (Equation (2)), and Ratio of Performance to Deviation (RPD) (Equation (3)) of the 10-fold-cross-validation:

$$R^2 = \frac{\sum_{i=1}^{n}(\hat{y}_i - \overline{y}_i)^2}{\sum_{i=1}^{n}(y_i - \overline{y}_i)^2} \tag{1}$$

$$\text{RMSE} = \sqrt{\frac{\sum_{i=1}^{n}\left(\hat{y}_i - y_i\right)^2}{n}} \tag{2}$$

$$\text{RPD} = \frac{\text{std}}{\text{RMSE}} \tag{3}$$

where $\hat{y}$ = predicted value, $\overline{y}$ = mean observed value, y = observed values, n = number of samples with i = 1, 2, ..., n, and the standard deviation of the observed values(std).

Thresholds for RPD can be found which classify the models into three categories: non reliable when RPD < 1.4, fair when 1.4 < RPD < 2 and excellent when RPD > 2 [45]. Minasny and McBratney, however, consider these thresholds to be arbitrary and based on no statistical or utilitarian basis [46]. We will, therefore, not use the thresholds as model performance indicators. The RPD values will be used for comparison to the literature.

### 2.5.3. CAI Thresholding

Several CAI thresholds (from 0.0 to 3.0 in steps of 0.25) were tested to exclude spectra affected by crop residue cover from the PLSR analysis. To ensure that the predictive accuracies of the several PLSR models based on various CAI thresholds are comparable, we must make sure that the training datasets are comparable. Vašát et al. have shown that training sets with larger variance achieve a more accurate prediction in terms of variance explained [47]. Therefore, Levene's test ("car" package in the R Core Team, 2017) was used to verify the assumption that variances are equal across all training sets based on the CAI threshold, with a significance level of α = 0.05. The SOC training sets partitioned by CAI thresholds were further analyzed by descriptive statistics and frequency histograms. The general overview of the modelling approach is displayed in Figure 4.

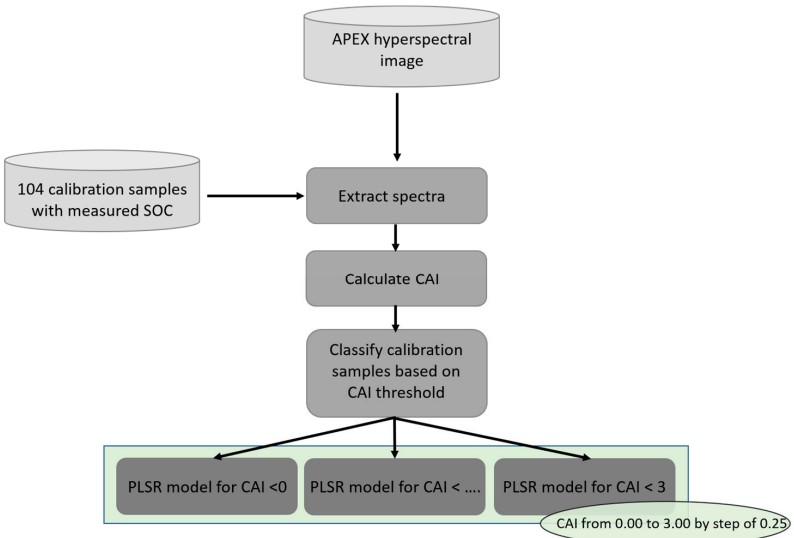

**Figure 4.** Overview of the soil organic carbon (SOC) prediction approach.

Additionally, to evaluate the possibility of S-2 MSI for SOC prediction, we used the hyperspectral airborne data to simulate the spectral bands of S-2. The resampling of airborne data according to the S-2 bands was conducted by means of the resample2 function in the "prospectr" package developed in R software. A PLSR model was then built on the resampled reflectance matrix in the same way as described in Section 2.5.2. The spatial resolution and the calibration dataset were kept constant.

### 2.6. Linking CAI and NBR2

To establish a link between CAI obtained from the hyperspectral sensor and NBR2 obtained from both hyperspectral and multispectral sensors, we randomly selected a set of 188 fields with an NDVI

below the threshold defined in Section 2.3 on both the S-2 (August) and the APEX image. We manually delineated these fields, while minimizing the border effect using a 60 m buffer. Then, 70 points (to reduce computation time) were randomly placed within each field for which the NBR2 index was calculated on the S2 images after bilinear interpolation. For the APEX image, first, the CAI index was calculated using a buffer of 10 m around each point. Then, we resampled the APEX spectra to the spectral resolution of S-2 MSI using the function resample2 in the "prospectr" package developed in R in order to calculate the NBR2 index.

### 2.7. Image Classification for Crop Residue Cover Quantification

A cloud free S-2A image was acquired on 13 October 2019. Then, fields were selected to cover the range in the crop residue variability, while avoiding vegetated areas (NDVI > 0.35 on the S-2 image). We took geo-referenced (7–8 m precision) RGB images (resolution of 3024 × 4032 pixels) of the surface of 59 fields from nadir with an iPhone 8 (18 October). The photos were then analyzed using RGB thresholds in order to estimate the residue cover. For each image a classification training set was manually selected. Then we used the maximum likelihood classification in ArcGIS 10.4 software by ESRI. If needed, the training sets creation was repeated until the end-result was judged satisfactory when comparing the classified image and the original RGB image. Overall, 43 images were adequately classified and were kept for further analysis. A value of 100% crop residue cover was associated to four images of maize fields at senescence.

## 3. Results

### 3.1. Quality of the APEX Spectra

The shape of APEX spectra of the reference surfaces is marked by features not shown on the ASD spectra (peaks of reflectance at 1000 nm and around 2050 nm; Figure 5A). These peaks are most probably artefacts due to the atmospheric conditions and the sensor characteristics. Remaining as close as possible to the original CAI bands and at the same time avoiding the artefacts of the APEX sensor, we selected the bands at 2026, 2100 and 2214 nm to be used in the remainder of the text (Figure 5B and Table 2). Moreover, spectra based on the ASD field spectrometer demonstrated a slightly different albedo intensity. Hence, the CAI values obtained from the APEX spectra are to be considered as relative, and should not be compared in absolute values to CAI values measured in the laboratory. This means that we cannot use results from Nagler et al. who established a relationship between CAI and crop residue cover [23].

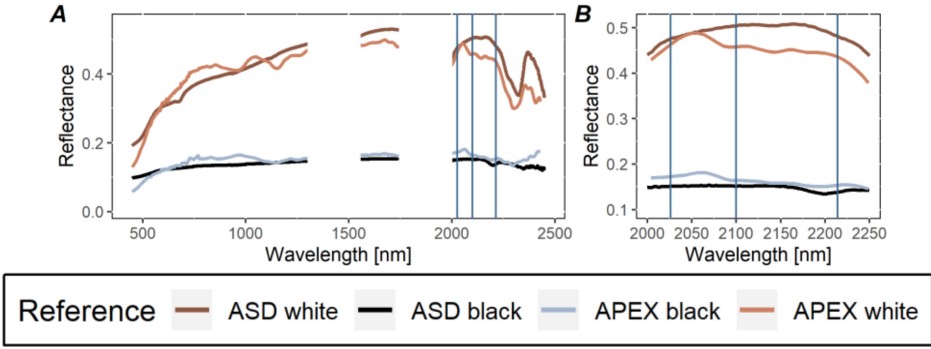

**Figure 5.** (**A**) Visible (Vis), near-infrared (NIR) and shortwave infrared (SWIR) (VNIR-SWIR) reflectance spectra of black and white references recorded by the ASD pro FR portable spectroradiometer and by the Airborne Prism Experiment (APEX) sensor; (**B**) zoom on the SWIR region where the soil spectra are the most affected by crop residues. The three vertical lines correspond to the cellulose absorption index wavelengths used in this study ($R_{2.0}$ = 2026 nm, $R_{2.1}$ = 2100 nm, $R_{2.2}$ = 2214 nm).

The average APEX spectrum acquired from 104 pixels where soil samples were collected showed a well-defined absorption feature at 2300 nm, produced by molecular vibrations related to SOC (Figure 6) [9]. The relationship between SOC and the spectral reflectance was mainly negative ($r < 0$). The expected negative relationship between SOC and reflectance is attributed to the stretching and bending of C–H, O–H and C=C and the electronic transition portions of the electromagnetic spectrum in the VNIR region [48].

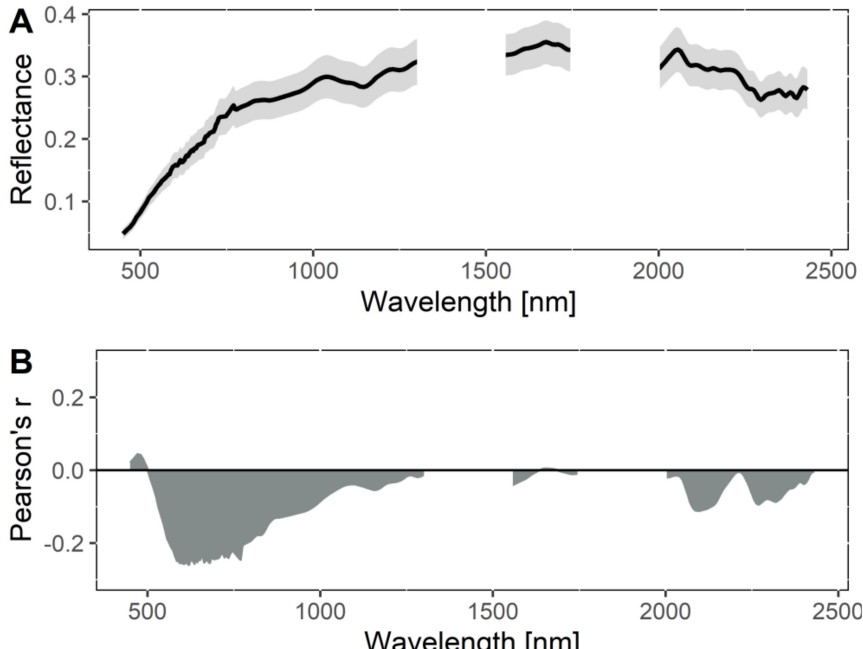

**Figure 6.** (**A**) Mean soil spectral signature (black line) and standard deviation (lower and upper boundaries) for the raw spectra of the 104 calibration samples extracted from the hyperspectral Airborne Prism Experiment (APEX) sensor (image from 2 September 2018) and (**B**) the Pearson's correlation coefficient of the soil organic carbon content with the raw hyperspectral reflectance.

### 3.2. SOC Prediction Models

### 3.2.1. Without CAI Threshold

We obtained the best SOC prediction using reflectance rather than absorbance (Table A1). The prediction accuracy for the ten-fold-cross validation yielded an $R^2$ of 0.49, an RMSE of 2.13 g kg$^{-1}$ and an RPD of 1.4 (Table A1, Figure 7A). VIP values larger than one indicate the wavelength that contributes to the model overall i.e., the visible (440–740 nm), the SWIR region (2000–2060 nm and 2380–2430 nm) and in particular, the blue and green regions with wavelengths between 450 and 550 nm (Figure 7B). The predicted SOC content was on average 12.1 g kg$^{-1}$ and was rather variable (variable coefficient (CV) = 24.5%, $n$ = 104, last line in Table 3).

The PLSR model for SOC content demonstrated sufficient accuracy to be applied on a pixel-by-pixel basis over the APEX image. For further model assessment, the model was firstly applied to the 276 sampling locations in three fields in the Belgian loam belt: Sicy, Hostellerie and Ernage. The three fields show a range in average CAI from 1.56 for Sicy, 0.52 for Hostellerie and 0.37 for Ernage (Figure 8). We observe a significant shift between the distribution of predicted and measured SOC contents in Sicy (Student's t-test at α = 0.05 ($p$-value = $2.2 \times 10^{-16}$); Figure 8A top panel), while such a shift is not observed for the other two fields (Hostellerie and Ernage; Figure 8B,C top panels). The scatterplot for Sicy with the highest CAI indicates an overestimation of the SOC values (Figure 8A bottom panel). Such overestimation is not observed for the fields with lower CAI values (Figure 8B,C bottom panels).

The crop residue cover of the Sicy field was estimated at 24% based on a supervised classification of an RGB image taken with an iPhone at the moment of APEX overflight.

**Table 3.** Descriptive statistics of the SOC calibration dataset and the results of the Partial Least Squares Regression (PLSR) model applying the Cellulose Absorption Index (CAI) thresholds. In bold is the best model. CV = variation coefficient, LV = latent variables.

| CAI Threshold | n | Min * | Max * | Mean * | Std * | CV (%) | LV | RMSE * | $R^2$ | RPD |
|---|---|---|---|---|---|---|---|---|---|---|
| | | | Descriptive Statistics | | | | | Tenfold-Cross-Validation | | |
| 0.00 | 40 | 7.5 | 16.5 | 10.9 | 2.1 | 19.5 | 3 | 1.98 | 0.14 | 1.07 |
| 0.25 | 58 | 7.5 | 16.5 | 11.2 | 2.2 | 19.8 | 4 | 1.84 | 0.32 | 1.20 |
| 0.50 | 75 | 7.5 | 19.9 | 11.5 | 2.6 | 22.4 | 6 | 1.87 | 0.48 | 1.38 |
| **0.75** | **80** | **7.5** | **19.9** | **11.6** | **2.8** | **23.0** | **13** | **1.75** | **0.59** | **1.51** |
| 1.00 | 90 | 7.5 | 19.9 | 11.7 | 2.8 | 23.4 | 4 | 2.11 | 0.40 | 1.29 |
| 1.25 | 95 | 7.5 | 20.0 | 11.8 | 3.0 | 24.0 | 4 | 2.21 | 0.39 | 1.29 |
| 1.50 | 99 | 7.5 | 20.2 | 12.0 | 3.0 | 25.0 | 13 | 2.31 | 0.45 | 1.30 |
| 1.75 | 100 | 7.5 | 20.2 | 12.0 | 3.0 | 24.9 | 5 | 2.33 | 0.40 | 1.28 |
| 2.00–2.50 | 102 | 7.5 | 20.2 | 12.0 | 3.0 | 24.7 | 12 | 2.26 | 0.45 | 1.32 |
| 2.75 | 103 | 7.5 | 20.2 | 12.1 | 3.0 | 24.7 | 14 | 2.25 | 0.47 | 1.32 |
| 3.00 | 104 | 7.5 | 20.2 | 12.1 | 3.0 | 24.5 | 13 | 2.13 | 0.49 | 1.39 |

\* expressed in g kg$^{-1}$.

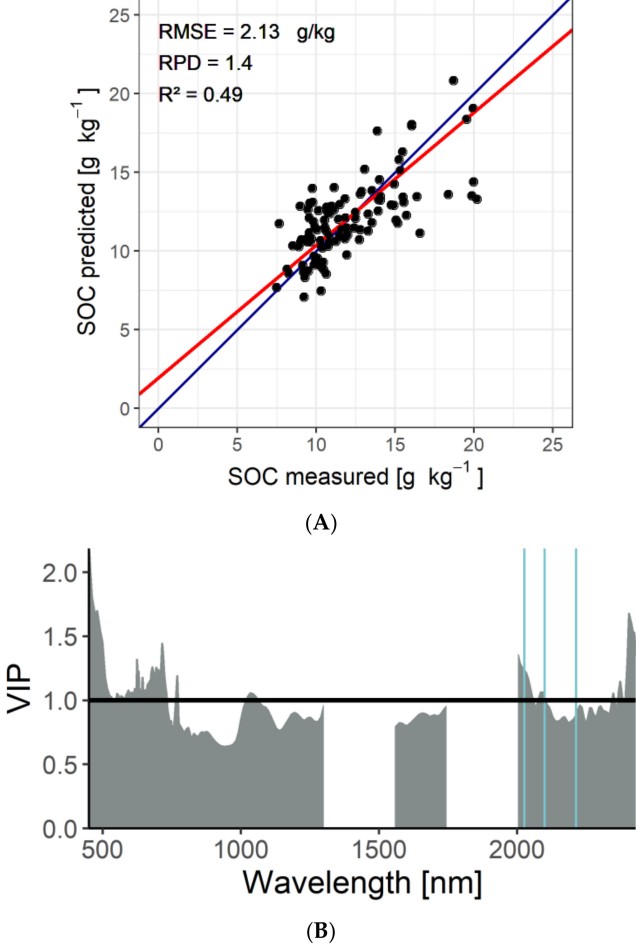

(A)

(B)

**Figure 7.** Measured against predicted soil organic carbon content (SOC) in the flight line of the hyperspectral Airborne Prism Experiment (APEX). The blue line is the 1:1 line, the red line is the regression line (**A**) with its variance importance in projection (VIP) scores (**B**). Predictors with VIP values greater than one (area above the horizontal line in the plot) were considered significant for the PLSR model. The blue vertical lines show the wavelengths used for the CAI index.

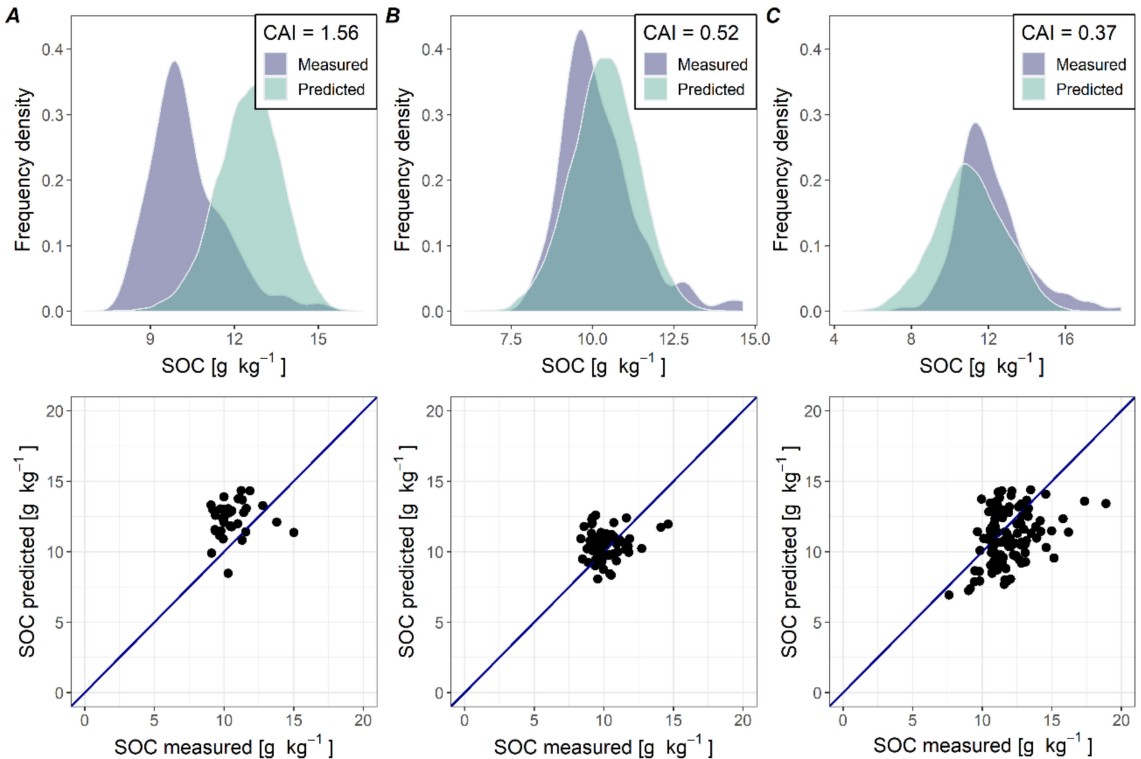

**Figure 8.** Comparison of the predicted SOC obtained from the Partial Least Squares Regression (PLSR) model and the measured SOC values in the (**A**) Sicy, (**B**) Hostellerie and (**C**) Ernage fields, and their respective Cellulose Absorption Index (CAI) values. The blue lines in the plots below are the 1:1 line. See Figure 1 for the location of the fields and Figure 2 for the location of the sampling points.

SOC and CAI maps developed from APEX data reveal very similar spatial patterns (Figures 9 and 10). An area of 18 croplands with an NDVI value below the 0.26 threshold was selected in the south-eastern part of the flight strip. The SOC map mainly shows an inter-field variability of SOC with values ranging from 6.8 to 14.8 g kg$^{-1}$. It appears that fields with higher predicted SOC correspond to those with higher CAI values. SOC and CAI maps of a single field confirm this observation (Geest, Figure 10). The Geest field appears to have a higher crop residue cover in the north-east (Figure 10B). The sharp and straight edges of this pattern point to farming practices such as ploughing and seeding rather than a natural gradient in residue cover reflected by the CAI. The same sharp-edged pattern is also found for the predicted SOC content. The overestimation in SOC in fields with a large CAI value (i.e., Sicy; Figure 8A) and the similar spatial patterns of CAI and SOC in the Geest field (Figures 9 and 10) strongly suggest that a partial residue cover leads to a positive bias in predicted SOC contents.

We fitted an exponential model to the observed semivariance of the predicted SOC contents of 8000 random points in an area of 30 km$^2$ in the south-eastern part of the flight strip (Figure 11, Table 4). The range is reached at 335 m, the same order of magnitude as an average field size.

### 3.2.2. With CAI Threshold

Several CAI thresholds (from 0.0 to 3.0 in steps of 0.25) were applied to create calibration subsets on which PLSR models were built to predict SOC content (Table 3). The CV shows a rather stable value for all subsets (19.5 for CAI < 0.00 and 25.0 for CAI < 1.50). The Levene's test indicates the homogeneity of variances across the calibration subsets (Table A2). All $p$-values were higher than the significance level of $\alpha = 0.05$, except for subset CAI < 0.00 against subsets CAI < 2.75 and CAI < 3.00. Those subsets have significantly different variances and should therefore not be compared. The model performance varies with the CAI threshold (Figure 12, Table 3): The lowest performance is

obtained if a strict threshold of CAI < 0.00 is applied. This is likely due to the low number of calibration samples ($n$ = 40). The model performance then increases to reach a maximum at CAI < 0.75 ($R^2$ = 0.59, RMSE = 1.75 g kg$^{-1}$ and RPD = 1.51). Then, model performance decreases and remains more or less stable for CAI thresholds from 1.00 to 3.00. The scatterplot for the measured and predicted SOC from absorbance spectra with a calibration subset of CAI < 0.75 is illustrated in Figure 13A. The dispersion of points is less marked than on the PLSR model without a CAI threshold (Figure 7A). Overall, similar patterns in VIP are observed with a peak in the blue and green regions regardless of the CAI threshold (Figures 7B and 13B).

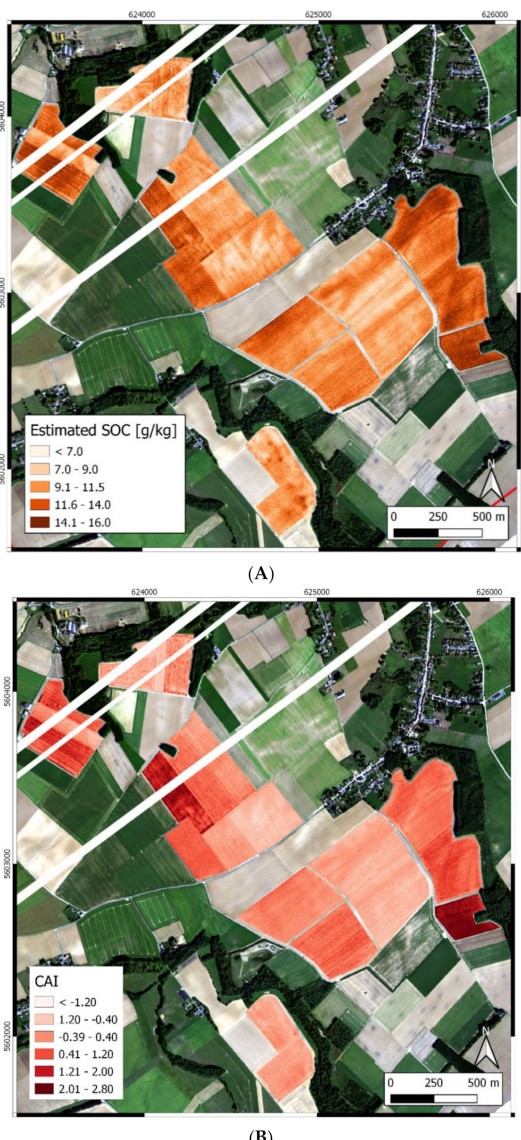

**Figure 9.** Comparison of the spatial patterns of (**A**) predicted SOC content by a Partial Least Squares Regression (PLSR) model on extracted airborne hyperspectral data and (**B**) the Cellulose Absorption Index (CAI) calculated from extracted airborne hyperspectral data in part of the APEX flight strip. The white stripes were removed because of noise in the airborne data.

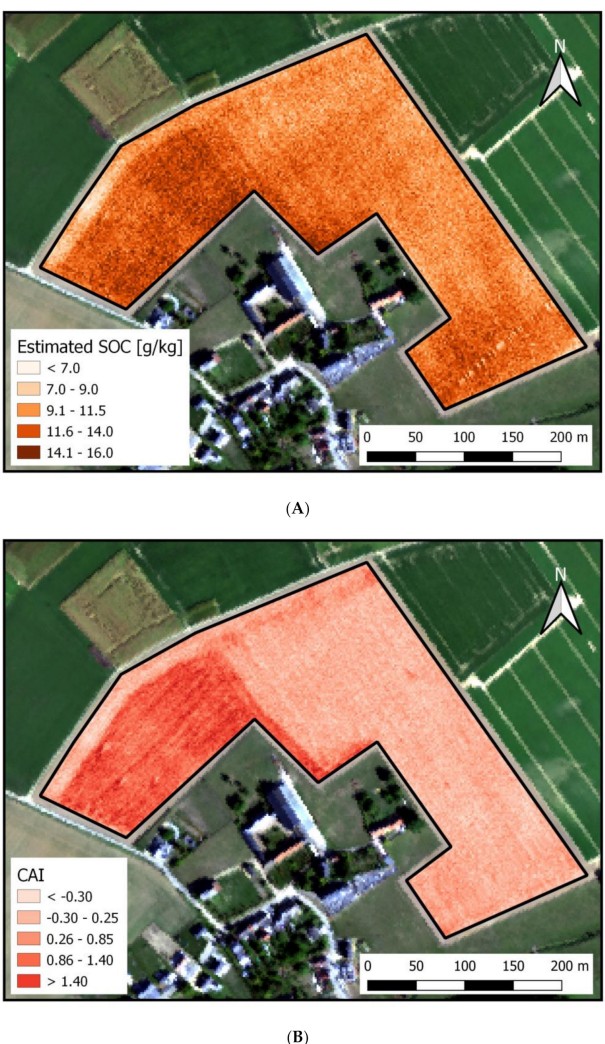

**Figure 10.** Comparison of the spatial patterns of (**A**) predicted SOC content by a Partial Least Squares Regression (PLSR) model on extracted airborne hyperspectral data and (**B**) the Cellulose Absorption Index (CAI) calculated from extracted airborne hyperspectral data in Geest field. For the location of the Geest field see Figure 1.

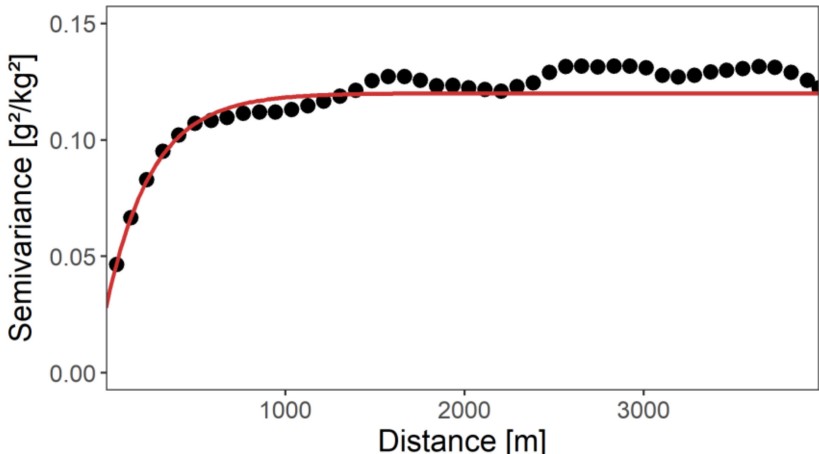

**Figure 11.** Variogram of the predicted SOC data for 8000 random locations. The points are the experimental semivariance (expressed in $g^2/kg^2$) values whereas the red line is the model of variogram.

**Table 4.** Variogram parameters of the fitted model (Figure 11) for predicted SOC obtained from a Partial Least Squares Regression (PLSR) model.

| Model | Sill (g$^2$/kg$^2$) | Range (m) |
|---|---|---|
| Nugget effect | 0.023 | |
| Exponential * | 0.109 | 335 |

* The exponential model has no finite range and here is reported the practical range. It is the distance at which the variogram value equals 95% of the sill variance [49].

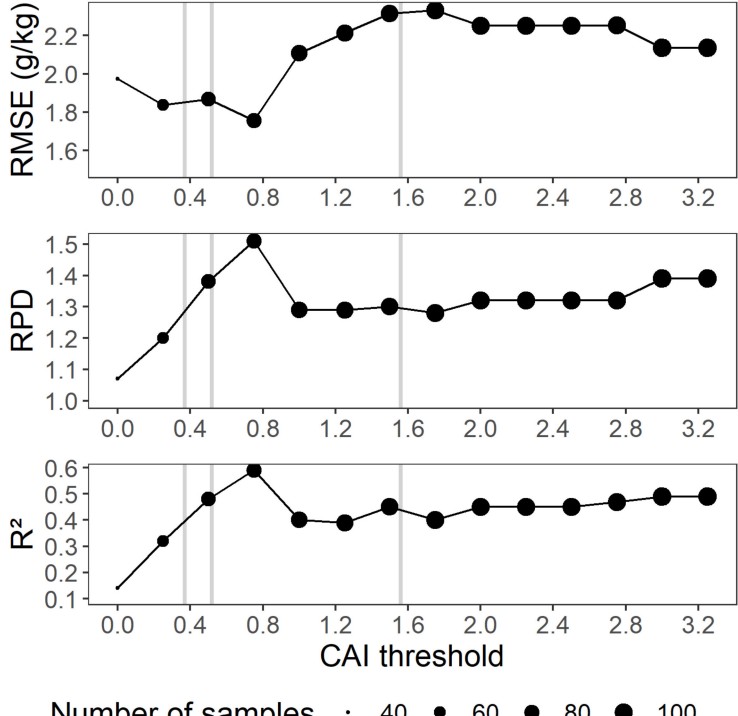

**Figure 12.** Coefficient of determination (R$^2$), Ratio of Performance to Deviation (RPD) and Root Mean Square Error (RMSE) of the Partial Least Squares Regression (PLSR) models for SOC content as a function of Cellulose Absorption Index (CAI) thresholds. The vertical lines indicate the average CAI values of three fields: Sicy: 1.56, Hostellerie: 0.52 and Ernage: 0.37 (Figures 2 and 8).

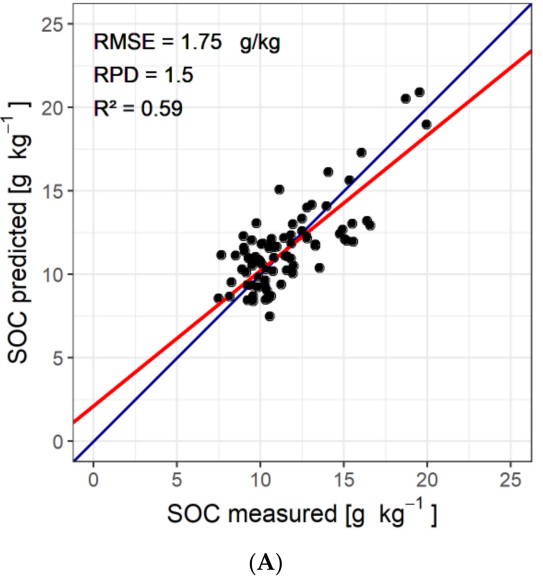

(**A**)

**Figure 13.** *Cont.*

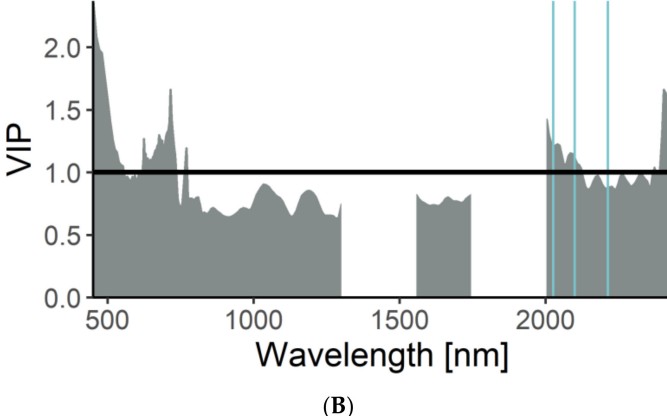

(**B**)

**Figure 13.** Measured against predicted soil organic carbon content (SOC) in the flight line of the hyperspectral Airborne Prism Experiment (APEX) sensor with a Cellulose Absorption Index (CAI) threshold at 0.75. The blue line is the 1:1 line, the red line is the regression line (**A**) with its variance importance in projection (VIP) scores (**B**). Predictors with VIP values greater than one (area above the horizontal line in the plot) were considered significant for the PLSR model. The blue vertical lines show the spectral wavelengths used for the CAI index.

The results of a PLSR model built on a reflectance matrix resampled to S-2 MSI spectral resolution are shown in Figure 14. The calibration subset was selected based on the CAI threshold that provided the best SOC prediction, i.e., CAI < 0.75. The model performance is inferior to that of the model built with hyperspectral data ($R^2$ = 0.48, RMSE = 1.91 g kg$^{-1}$ and RPD = 1.4).

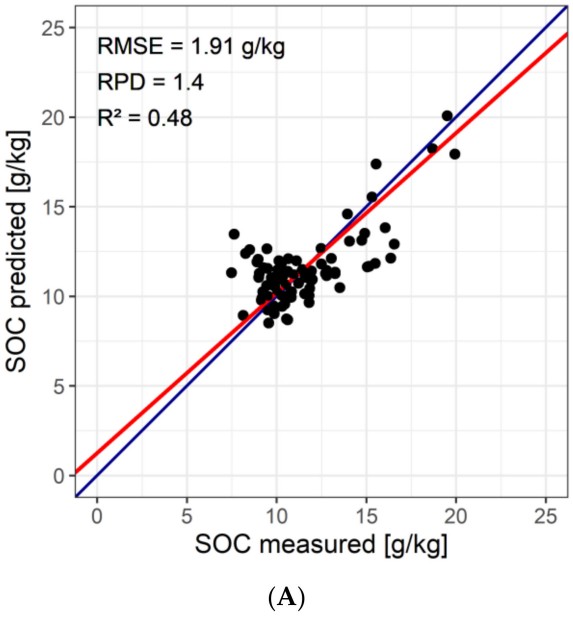

(**A**)

**Figure 14.** *Cont.*

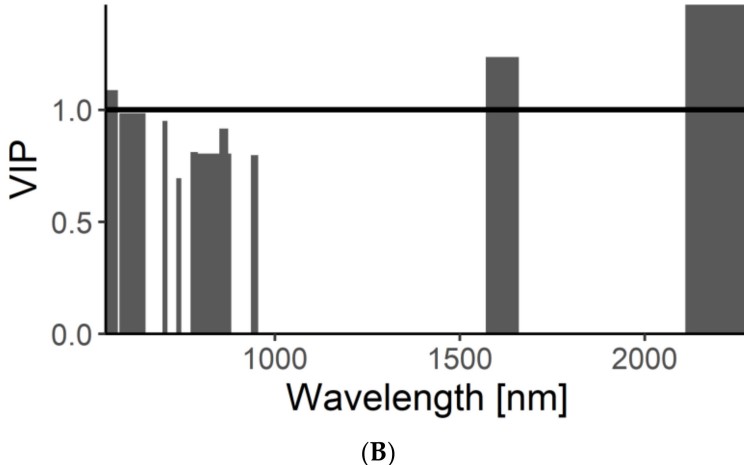

(**B**)

**Figure 14.** Measured against predicted soil organic carbon content (SOC) in the flight line of the hyperspectral Airborne Prism Experiment (APEX) sensor with a Cellulose Absorption Index (CAI) threshold at 0.75, where spectra were resampled to Sentinel-2 radiometric resolution. The blue line is the 1:1 line, the red line is the regression line (**A**) with its variance importance in projection (VIP) scores (**B**). Predictors with VIP values greater than one (bars reaching above the horizontal line in the plot) were considered significant for the PLSR model.

### 3.3. Crop Residue Detection Using Sentinel-2

The CAI and NBR2 values both calculated on APEX spectra from the same date show a very good linear relationship ($R^2$ = 0.83; Figure 15A, Table 5). The NBR2 index calculated on S-2 MSI and on resampled APEX hyperspectral data also shows a linear relationship (Figure 15B). However, a cloud of points with a clearly higher NBR2 recorded on the S-2 images occurs (shaded area in Figure 15B). Moreover, the relationship does not follow the 1:1 line (Figure 15B). Therefore, the absolute values of NBR2 obtained from different platforms cannot be compared. The points in the blue shaded area of Figure 15B,C have high NBR2 values on the S-2 image from 24 August 2018, but lower NBR2 values on the APEX image nine days later. Hence, as high NBR2/CAI values are linked to extensive crop residue cover [25,34], we hypothesize that those are fields that have been ploughed during the nine day interval between the S-2 and the APEX images acquisition.

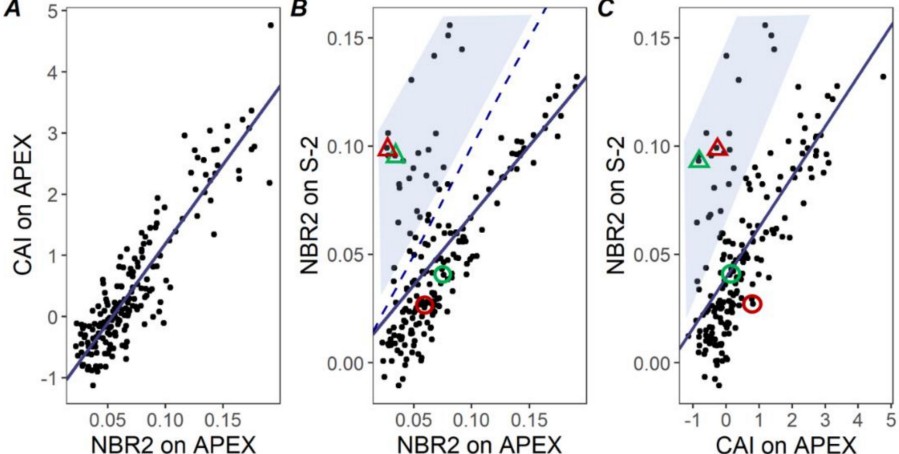

**Figure 15.** Relationships between the Cellulose Absorption Index (CAI) and the Normalized Burn Ratio 2 (NBR2) calculated on airborne hyperspectral data (APEX image from 2 September 2018) and on satellite multispectral data (S-2 image from 24 August 2018) on a set of 188 fields randomly selected in the Belgian loam belt (**A**–**C**). The dashed line in (**B**) represents the 1:1 line. The blue lines represent the linear regression (Table 5). For the explication of the blue shaded area, please see Section 3.3. The red symbols represent fields from Figure 16A and the green symbols from Figure 16B.

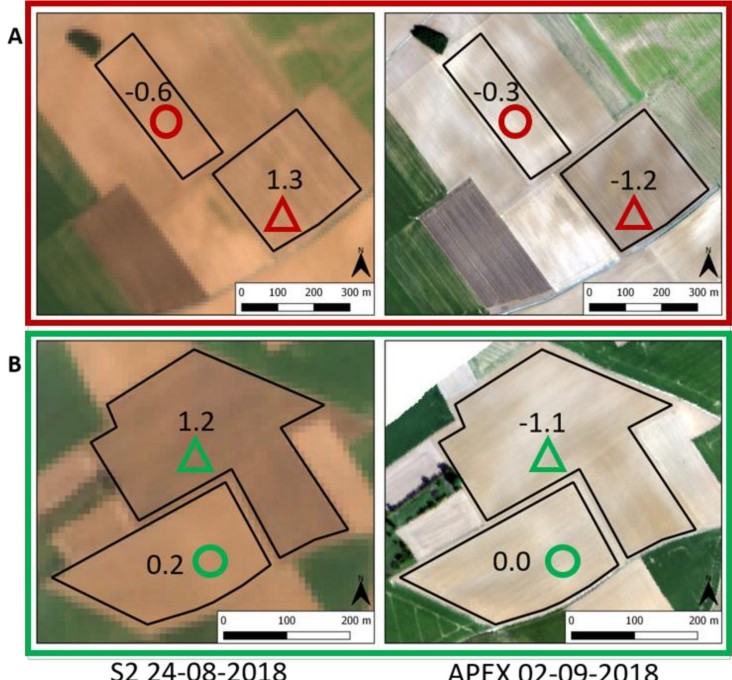

**Figure 16.** RGB images of pairs of selected fields (**A**,**B**) in the Belgian loam belt acquired by the Multi-spectral Instrument (MSI) Sentinel-2 sensor on 24 August 2018 (red: 665 nm, green: 560 nm, blue: 490 nm) (left) and the hyperspectral Airborne Prism Experiment (APEX) sensor on 2 September 2018 (red: 616 nm, green: 551 nm, blue: 470 nm) (right). The numerical values are the standardized NBR2 values extracted from Sentinel-2 MSI and resampled APEX sensor. The red (**A**) and green (**B**) circles represent fields which follow the regression lines on Figure 15B,C, while the red (**A**) and green (**B**) triangles represent fields which do not follow the regression line, and which have likely been ploughed.

**Table 5.** Linear regression models for mean values of the Cellulose Absorption Index (CAI) and the Normalized Burn Ratio 2 (NBR2) of a field calculated on airborne hyperspectral data (APEX image from 2 September 2018) and satellite multispectral data (S-2 image from 24 August 2018).

|  |  | Coefficients | $R^2$ | n |
|---|---|---|---|---|
| CAI (APEX)~NBR2 (APEX) | Intercept | −1.403 *** | 0.83 | 188 |
|  | Slope | 25.955 *** |  |  |
| NBR2 (APEX)~NBR2 (S-2) | Intercept | 0.003 *** | 0.43 | 188 |
|  | Slope | 0.642 *** |  |  |
| CAI (APEX)~NBR2 (S-2) | Intercept | 0.039 *** | 0.47 | 188 |
|  | Slope | 0.023 *** |  |  |

The significance levels are given as *** $p$-value $\leq$ 0.001.

To strengthen our hypothesis, RGB images of two examples of neighboring fields are provided for visual inspection (Figure 16). The red symbols (circle and triangle) on Figure 15B,C are the values in the fields in Figure 16A and the green symbols in Figure 15B. In each pair of RGB images, the color of one field does not change during the interval between the S-2 and APEX image (noted by a circle), while the color of the other field changes (noted by a triangle). The fields in which the color changes are those situated in the blue shaded areas in Figure 15B,C. Based on standardized NBR2 values, the NBR2 changes from 1.3 to −1.2 (i.e., 2.5 units) for the field in the blue shaded area which color has changed, while for the unchanged fields, they change from −0.6 to −0.3 (i.e., −0.3 units; Figure 16A). Similar trends can be observed for the fields in Figure 16B.

In addition, we provide the preliminary results of a field experiment, where we test the potential of the NBR2 index based on S-2 multispectral data for estimating the crop residue cover during wet conditions in autumn. Pixels with an NBR2 comprised between 0.04 and 0.11 show no discernable

trend with crop residue cover, which is lower than <15%. Pixels with a higher NBR2 (but a smaller range is observed: from 0.11 to 0.16), correspond to crop residue cover between 15% and 100%.

## 4. Discussion

We have demonstrated that APEX airborne imagery is capable of predicting SOC content over a relatively large area of 240 km$^2$ dominated by croplands for which the soil surface conditions were not controlled (R$^2$ = 0.49, RMSE = 2.13 g kg$^{-1}$ and RPD = 1.39). The RPD might seem relatively low compared to other similar studies which reported RPD values up to 2.7 for an area in northern Germany [8] and 2.39 for a north–south flight strip in Luxembourg [37]. The low RPD in the Belgian loam belt is mainly due to the small variation in SOC content (Table 3). Castaldi et al. highlighted the necessity of a suitable strategy for the calibration dataset i.e., covering both the spectral variability of the area and at the same time the full range of SOC values [8]. Considering the variability in SOC content, the PLSR model yields similar performances as reported by Castaldi et al. [8] and Shi et al. [50] for the same region. The SOC prediction maps mainly show a pattern of SOC variability between fields (Figures 9A and 10A). This pattern was confirmed by the semivariogram that yields a range of 335 m corresponding to the average dimension of the fields (Figure 11). These results might lead to a conclusion that the region mainly shows variability between fields, likely related to differences in land management or land use history. Yet, when the predicted SOC distribution of three fields was compared to the measured one, we observed a significant overestimation of SOC content for the field with a high CAI index (1.56, Sicy; Figure 8). This overestimation was neither observed for the Hostellerie with a CAI of 0.52 nor for the Ernage field with a CAI of 0.37. Hence, the variation in SOC between fields can to some extent be attributed to the differences in residue cover during the period where fields are being prepared for seeding winter cereals.

We therefore tested different CAI thresholds for selecting the calibration samples and found an optimum at CAI = 0.75 (Table 3, Figure 12). When using a more restrictive CAI threshold (CAI < 0.75), the accuracy decreased (R$^2$ dropped from 0.59 to 0.14, while RMSE increased by 0.23 g kg$^{-1}$), most likely due to the low number of calibration samples. Debaene et al. found that a minimum of 79 well distributed calibration samples is required for building a good predictive model for SOC content [51]. Hence, the optimum CAI threshold might represent a trade-off between a sufficient number of calibration samples and the purity of the pixels used for a SOC prediction model and is therefore case specific. Moreover, when a threshold of CAI < 0.75 was applied to the calibration samples, the comparison between resampled airborne data according to S-2 bands and the hyperspectral APEX data did not show a remarkable difference in terms of SOC prediction accuracy (R$^2$ = 0.59, RMSE = 1.75 g kg$^{-1}$ and RPD = 1.51 for airborne hyperspectral versus R$^2$ = 0.48, RMSE = 1.91 g kg$^{-1}$ and RPD = 1.40 for resampled to S-2 spectral resolution). Similar results were obtained by Castaldi et al. [8].

Additionally, we tested the link between CAI and NBR2. During the time of the overflight of both the S-2 and APEX sensors the soil surface was dry, as no rainfall preceded these days (Figure 3A) and the surface quickly dries out during the warm summer conditions. Moreover, soil roughness was generally low because fields were either in seedbed conditions or had just been harvested. The main disturbing factor was therefore the crop residue partially covering the soil surface. In such conditions, the CAI and NBR2 calculated on both S-2 multispectral data and resampled APEX hyperspectral data showed a linear relationship (Figure 15). Therefore, if we take into account the linear relationship between crop residue cover and CAI as found by Nagler et al. [23], we can conclude that in dry conditions and on relatively smooth soils, both the CAI and NBR2 show a linear relationship with crop residue cover. However, we have illustrated that CAI and NBR2 values obtained by various platforms (laboratory, airborne or spaceborne) cannot be compared in absolute values, mainly due to atmospheric disturbance.

In order to test whether the linear relationship between NBR2 and crop residue holds when soil conditions are different, we collected photos of the soil surface in mid-autumn, after a period of heavy rainfall, and compared the residue cover to the NBR2 index. We illustrated that in such moist soil

conditions, the relationship between NBR2 and crop residue cover is poor (Figure 17). Pixels with a variation of 0.04 to 0.11 in NBR2 had less than 15% crop residue cover, while pixels with higher NBR2 values but covering a smaller range (0.11 to 0.16) had between 15% and 100% crop residue cover. This might indicate that moist soil conditions mask the effects of crop residue cover, and/or that the effects of crop residues and soil moisture are both reflected by the NBR2 index, as has already been suggested by Musick and Pelletier [52] and Quemada and Daughtry [53]. Indeed, Haubrock et al. defined the Normalized Difference Moisture Index (NMSI) as an indicator of soil moisture content combining reflectance values at 1800 and 2119 nm [54]. The NBR2 index has one band at the shoulder of the water absorption band, coinciding with the NMSI band at 2119 nm (Figure 18). The reflectance values in S-2 bands 11 and 12 for dry, wet, and dry soil spectra with 24% residue cover (Figure 18) show that the difference in band 11 and band 12 is similar for wet soil (black curve in Figure 18) and dry soil with crop residues (orange curve in Figure 18). Hence, this combined effect of crop residues and soil moisture on the NBR2 index does not allow for a correct selection of pure soil pixels suitable for SOC prediction. Yue et al. proposed a broadband spectral angle index (BAI), which corrects for the effects of soil moisture and crop residue cover on spectral reflectance [55]. While working on small plots, they have successfully managed to quantify maize residue cover on soils of variable moisture, using BAI on S-2 data. It remains yet to be tested if the methodology of Yue et al. can be applied to larger areas with various crop residue types.

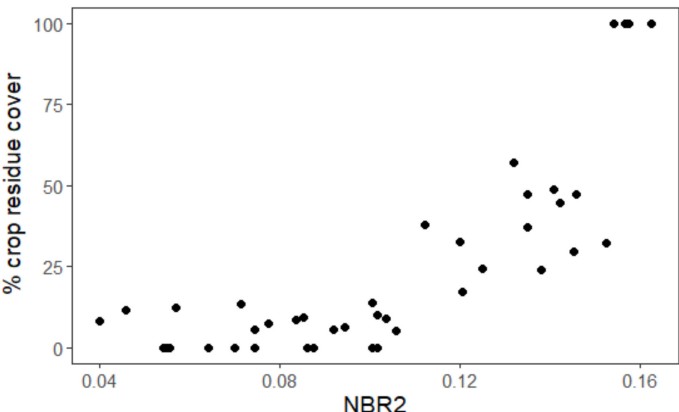

**Figure 17.** Normalized Burn Ratio 2 (NBR2) calculated on S-2 MSI spectra from 13 October, 2019 plotted against crop residue cover for 59 samples.

Vaudour et al. demonstrated the impact of acquisition date on the prediction performance, as surface conditions vary during the growing season in function of the crop rotation [16]. Cierniewsky and Ceglarek estimated the annual bare soil distribution throughout the year in the most extensive agricultural regions, and found that for western Europe the highest amount of bare croplands, and therefore the optimal window to build SOC prediction models, occurs in mid-April and late summer to the beginning of September [56]. In late summer, cereals have been harvested and the bare soils are likely to be dry, with stubbles on the fields that have not yet been seeded. In mid-April, some fields that have been ploughed in autumn have developed a surface crust, while other fields are seeded with summer crops. Hence, the acquisition date is a first indicator of the soil surface conditions, which might be encountered in an agricultural region, and thereby of the challenges to isolate pixels mimicking a soil sample as scanned in the laboratory. In this study, we have shown that in late summer, it is the crop residue effect that induces the largest error in the SOC prediction models. A similar approach should therefore be tested for the effects of soil crusts on SOC prediction models in mid-April.

Recently, many studies have focused on the use of existing datasets, such as the Land Use/cover Area frame Survey (LUCAS) topsoil spectral library, for estimation of SOC [7,44,57]. Such large available soil databases allow the prediction of SOC at a low-cost with reasonable accuracy. The best results are obtained using a local regression approach where calibration samples are selected based

on their spectral similarity, [57]. We believe that progress in SOC mapping can be made using (i) a local modelling approach for large-scale spectral libraries and (ii) by choosing imagery acquired when the soils are, as much as possible, in the same condition as laboratory samples (i.e., smooth, dry and without residues or surface crusts).

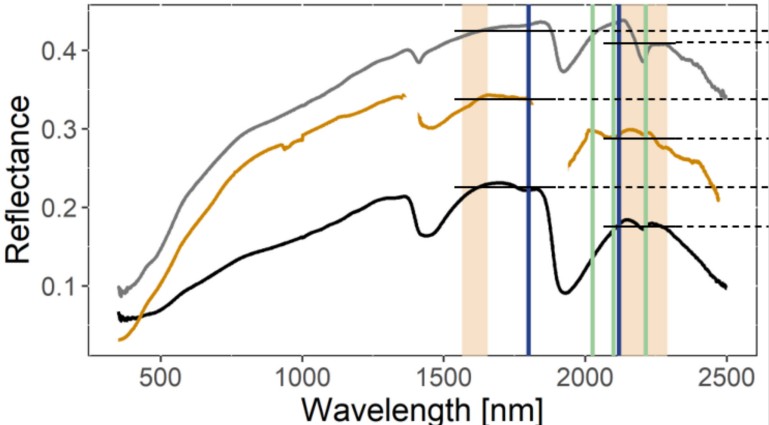

**Figure 18.** Laboratory reflectance spectra of a dry soil sample (gray line) and a wet soil sample (black line, 25% water content), and ASD spectra of a field with 24% residue cover (orange line). The blue vertical lines indicate the position of the Normalized Soil Moisture Index (NSMI) and the green vertical lines the position of the Cellulose Absorption Index (CAI). The beige shadowed areas are the two Sentinel-2 bands 11 and 12 used for the Normalized Burn Ratio (NBR2). The black horizontal lines represent the gap between the reflectance values in the Sentinel-2 bands 11 and 12 for the three spectra.

## 5. Conclusions

We have shown that when SOC prediction models are applied to a large area without an a priori knowledge of soil surface conditions, the disturbing effects of crop residues influence the SOC prediction accuracy. In particular, an overestimation of SOC has been observed for fields with extensive residue cover. We then tested the effect of the threshold for a hyperspectral index linked to crop residues (CAI) on the performance of SOC prediction models in Belgian loam belt croplands. For dry soils in seedbed condition, the pure pixel selection based on CAI thresholds improves the SOC prediction accuracy. In particular, a CAI threshold of 0.75 allowed for the best SOC prediction model.

Additionally, we have demonstrated that when soils are dry and in seedbed condition, CAI and NBR2 indexes based on both hyperspectral airborne and multispectral satellite sensors show a linear relationship. By extrapolation, a linear relationship exists between crop residue cover and both CAI and NBR2. However, we also found that the linear relationship between NBR2 and crop residue cover does not hold when soils are wet.

**Author Contributions:** Conceptualization, K.D. and B.v.W.; methodology, K.D. and B.v.W.; software, K.D.; validation, K.D., B.v.W., P.S.; formal analysis, K.D.; investigation, Q.L., P.S. and K.D.; resources, B.v.W. and Q.L.; data curation, Q.L., B.v.W. and K.D.; writing—original draft preparation, K.D.; writing—review and editing, K.D., B.v.W., P.S.; visualization, K.D.; supervision, B.v.W.; project administration, B.v.W.; funding acquisition, B.v.W. All authors have read and agreed to the published version of the manuscript.

**Funding:** This research was funded by the Belgian Federal Science Policy Office (BELSPO) as part of the UAVSOIL project "UAV borne spectrometers for high resolution soil and crop monitoring" (Contract SR/00/362). The APEX imagery was acquired within the STEREO III Belair program of the Belgian Federal Science Policy Office (BELSPO) and the support is gratefully acknowledged.

**Acknowledgments:** We thank Marco Bravin of the Earth and Life Institute of the Université Catholique de Louvain (UCLouvain) for the essential organic carbon measurements. We thank the VITO Remote Sensing Department for providing the APEX images.

**Conflicts of Interest:** The funders had no role in the design of the study; in the collection, analyses, or interpretation of data; in the writing of the manuscript, or in the decision to publish the results.

## Appendix A

**Table A1.** Descriptive statistics of the SOC calibration dataset and the results of the Partial Least Squares Regression (PLSR) model per various spectral pre-treatments. R = Reflectance, A = Absorbance, SG = Savitzky–Golay smoothing, FD = 1st derivative, SD = 2nd derivative, SNV = standard normal variate, CR = continuum removal. In bold is the pre-treatment which provides the best model.

| Treatment | LV | RMSE * | $R^2$ | RPD |
|---|---|---|---|---|
| Reflectance | 7 | 2.40 | 0.38 | 1.24 |
| **Absorbance** | **13** | **2.14** | **0.49** | **1.39** |
| SG-R | 7 | 2.42 | 0.36 | 1.23 |
| SG-A | 9 | 2.33 | 0.40 | 1.28 |
| FD | 7 | 2.26 | 0.44 | 1.32 |
| SG-FD | 12 | 2.36 | 0.39 | 1.26 |
| SD | 9 | 2.80 | 0.24 | 1.06 |
| SG-SD | 12 | 2.31 | 0.41 | 1.28 |
| SNV | 5 | 2.45 | 0.33 | 1.21 |
| SNV detrend | 4 | 2.40 | 0.36 | 1.24 |
| CR | 6 | 2.29 | 0.42 | 1.29 |

* expressed in g kg$^{-1}$.

**Table A2.** *p*-value of Levene's test of homogeneity of variances between training sets of various CAI thresholds. Null hypothesis: the population variances are equal. In bold: training set combinations that reject the null hypothesis at $\alpha = 0.05$ and whose variances are therefore not equal.

| CAI | 0.00 | 0.25 | 0.50 | 0.75 | 1.00 | 1.25 | 1.50 | 1.75 | 2.00 | 2.25 | 2.50 | 2.75 |
|---|---|---|---|---|---|---|---|---|---|---|---|---|
| 0.00 | - | | | | | | | | | | | |
| 0.25 | 0.532 | - | | | | | | | | | | |
| 0.50 | 0.312 | 0.598 | - | | | | | | | | | |
| 0.75 | 0.253 | 0.491 | 0.871 | - | | | | | | | | |
| 1.00 | 0.195 | 0.383 | 0.726 | 0.852 | - | | | | | | | |
| 1.25 | 0.114 | 0.219 | 0.465 | 0.565 | 0.69 | - | | | | | | |
| 1.50 | 0.059 | 0.106 | 0.248 | 0.313 | 0.396 | 0.65 | - | | | | | |
| 1.75 | 0.063 | 0.115 | 0.270 | 0.341 | 0.430 | 0.696 | 0.946 | - | | | | |
| 2.00 | 0.057 | 0.107 | 0.260 | 0.330 | 0.420 | 0.689 | 0.948 | 0.997 | - | | | |
| 2.25 | 0.057 | 0.107 | 0.260 | 0.330 | 0.420 | 0.689 | 0.948 | 0.997 | 1.000 | - | | |
| 2.50 | 0.057 | 0.107 | 0.260 | 0.330 | 0.420 | 0.689 | 0.948 | 0.997 | 1.000 | 1.000 | - | |
| 2.75 | **0.049** | 0.092 | 0.230 | 0.294 | 0.377 | 0.634 | 0.994 | 0.938 | 0.940 | 0.940 | 0.940 | - |
| 3.00 | **0.046** | 0.087 | 0.223 | 0.287 | 0.369 | 0.627 | 0.99 | 0.934 | 0.936 | 0.936 | 0.936 | 0.997 |

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
