# Peer review of "Soil Organic Carbon Mapping from Remote Sensing: The Effect of Crop Residues"

_remotesensing, doi:10.3390/rs12121913_

Round 1

Reviewer 1 Report

This is interesting research about the mapping of Soil organic carbon using remote sensing. The manuscript needs to be revised according to the comments/suggestions made below before considering it for publishing.

The authors need to provide some figures to:
Figure 1. The map is too simplistic. The authors should add which coordinate system (projection, ellipsoid, and datum) is used here (see the example below). See here http://tiny.cc/91kynz
Please include all the spatial reference properties see the example below:
Projected Coordinate System: WGS_1984_UTM_Zone_21N
Projection: Transverse Mercator
Linear Unit: Meter
Geographic Coordinate System: GCS_WGS_1984
Datum: D_WGS_1984
Prime Meridian: Greenwich
Angular Unit: Degree

In the "Materials and Methods" section (this section should be reorganized). I strongly recommend to the authors devise a flowchart that depicts the steps that they have processed in this study.

Reviewer 2 Report

In the introduction, a clearer picture of remote sensing techniques could be given:

  • Explain better what is remote sensing techniques and what are the techniques.
  • What is Airborne Prism Experiment? And what are the advantages or disadvantages?
  • Row 47-49. Were the same remote sensing techniques used in the cited bibliography used?

Line 113-134. Explain why two sets of samples were taken with different surfaces.

Were the set of 276 samples taken in the same period as the 104 sample set?

Line 349-351. The sentence is not very clear:

“The SOC content was on average 12.1 g Kg-1 and was rather variable with a variation coefficient (CV) of 24.5% for n=104 (Table 3). The best SOC prediction was obtained when reflectance was transformed into absorbance (Table A1). Indeed, the prediction accuracy for….”

Line 371. Could the p-value be specified?

Line 373-375. Could one estimate how much SOC is overestimated?

Why is only CAI value of the Sicy field so high?

Line 426. How many samples are there with a CAI value < 0.75? Is the number of sample Always 40?

Line 529-521.  Also specify in rows 239-240-241 that the acquisitions were made after a period of heavy rainfall.

Reviewer 3 Report

The topic of this manuscript is of interest and well written. I liked reading it. Overall it is clear that the authors have collected a significant amount of data in the field, and the study, if correct, comes to some valuable conclusions. Therefore, I have one comment and suggestion:

In this study, SOC has been measured (estimated ) using both in situ and remote sensing data! Did the authors try to use other external SOC datasets such as (LUCAS topsoil) to see and evaluate how the feasibility of the selected approaches in this study on other datasets (LUCAS)! I think worth a try! LUCAS topsoil dataset consists of  20 thousand soil samples, including SOC and their hyperspectral measurements! Furthermore, it is important to test and evaluate the possibility of measuring SOC using satellite imagery such as (Sentinel-2). In this case, we would suggest to simulate the spectral bands from the Airborne Prism Experiment (APEX) to the spectral bands of sentinel-2 and explore the possibility of estimating SOC from simulated data.

Round 2

Reviewer 2 Report

The authors responded fully to all comments